# Analysis of the Immunogenicity of African Swine Fever F317L Protein and Screening of T Cell Epitopes

**DOI:** 10.3390/ani14091331

**Published:** 2024-04-29

**Authors:** Ying Huang, Wenzhu Zhai, Zhen Wang, Yuheng He, Chunhao Tao, Yuanyuan Chu, Zhongbao Pang, Hongfei Zhu, Hong Jia

**Affiliations:** 1Institute of Animal Sciences, Chinese Academy of Agricultural Sciences, Beijing 100193, China; hy811cysbm@163.com (Y.H.); shandixhqm@163.com (W.Z.); wz2893963594@126.com (Z.W.); heyuheng2012@163.com (Y.H.); chunhao_tao@163.com (C.T.); 18522961671@163.com (Y.C.); 82101205453@caas.cn (Z.P.); bioclub@vip.sina.com.cn (H.Z.); 2Animal Science and Technology College, Beijing University of Agriculture, Beijing 100193, China

**Keywords:** African swine fever virus, F317L protein, cellular immunity, humoral immunity, T cell epitope

## Abstract

**Simple Summary:**

The African swine fever virus (ASFV) is a linear double-stranded DNA virus characterized by a complex immune escape mechanism that has caused substantial economic losses to the global swine industry. An in-depth study of ASFV has revealed that screening for protective antigens and activating cellular immunity is essential for the development of ASF vaccines. In this study, we sought to examine the immunogenicity of the F317L protein of ASFV in mice while also identifying potential active T-cell epitopes in order to provide a basis for the functional study of the F317L protein, in addition to a reference for the subsequent development of ASF vaccines.

**Abstract:**

The African swine fever virus (ASFV) encodes numerous proteins characterized by complex immune escape mechanisms. At present, the structure and function of these proteins, including the F317L protein, have yet to be fully elucidated. In this study, we examined the immunogenicity of the F317L protein. Mice were subcutaneously immunized with the F317L protein using initial and subsequent booster doses, and, at the 28th day post-treatment, we assessed the humoral and cellular immune responses of mice. The F317L protein stimulated production of specific antibodies and activated humoral immune responses. In addition, F317L stimulated the production of large amounts of IFN-γ by splenic lymphocytes, thereby activating cellular immune responses. Using informatics technology, we predicted and synthesized 29 F317L protein T cell epitopes, which were screened using IFN-γ ELISpot. Among these, the F25 (^246^SRRSLVNPWT^255^) peptide was identified as having a stronger stimulatory effect than the full-length protein. Collectively, our findings revealed that the ASFV F317L protein can stimulate both strong humoral and cellular immunity in mice, and that the F25 (^246^SRRSLVNPWT^255^) peptide may be a potential active T cell epitope. These findings will provide a reference for further in-depth studies of the F317L protein and screening of antigenic epitopes.

## 1. Introduction

African swine fever (ASF) is an acute highly contactable infectious disease caused by the African swine fever virus (ASFV) that infects domestic and wild pigs [1]. Highly virulent strains of the virus can cause hyperacute and acute disease, resulting in 100% mortality in pigs within a short period of time after infection [1,2]. ASF is classified as a legally reportable animal disease by the World Health Organization, and it is also classified as a Class I animal disease in China. Since the first reported outbreak of ASF in Shen-yang City, Liaoning Province, China, in August 2018, ASF has spread rapidly to most parts of China, causing considerable economic losses to the domestic pig farming industry.

The relatively large ASFV genome encodes between 150 and 200 proteins [3]. The virion consists of five concentric layers, extending outward from the central inner core (via the inner core shell, inner membrane, and capsid) to the exterior vesicle, each of which comprises distinct protein types [4,5]. These mainly include structural and immunoregulatory proteins [6], including the inner membrane proteins p54, p17, and p30; coat proteins p72 and p49; and vesicle proteins CD2v and p12. Among these, it has previously been confirmed that the ASFV p54, p72, and p30 proteins can induce the production of neutralizing antibodies in experimental animals [7,8]. Attack protection experiments revealed that immunizing pigs with these three proteins failed to protect these pigs, indicating that the induced production of neutralizing antibodies was insufficient in regard to conferring immunity [9].

The findings of further in-depth studies on the virus have revealed that the speed and efficacy of the neutralizing antibody response show considerable variation [7]; additionally, there may be differences in the neutralization mechanism of the antibodies elicited by different proteins. Consequently, the associations between the neutralizing antibodies and immunity protection against ASFV are in need of further experimental verification [10,11]. In this regard, it has been established that ASFV specific antibodies alone are insufficient for protecting against ASFV infection, while CD8^+^ lymphocyte subsets have been demonstrated to play an important role in protective immunity against ASFV [12,13]. Furthermore, a close association between cellular immune responses and protective responses has been identified in inbred pigs [14], and Argilaguet et al. [15] have described a DNA vaccination strategy in which 33% of the immunized pigs were immunoprotected, with this protective effect being attributed to an induction of CD8^+^ T cells. The findings of this study thus indicate that T cell immunity plays an important role in immunity protection. However, while, at present, ASFV protective antigens have yet to be identified, given that peptides induce protective responses, further screening and characterization of antigenic T-cell epitopes [16] are considered to be essential for the development of effective ASF vaccines.

On the basis of a literature search, we found that the late F317L protein of ASFV has been demonstrated to have immunosuppressive effects by impairing activation of the NF-kB pathway, thereby disrupting NF-kB activity [17]. In a further study, the authors analyzed the reactogenicity of the F317L protein and predicted that it has six B-cell antigenic epitopes [18]. The findings of such studies have thus provided evidence to indicate that ASFV F317L is a potential immunogenic antigen. Consequently, in this study, we sought to analyze F317L, and, accordingly, found that this protein can elicit both humoral and cellular immunity in experimental mice. In addition, we performed in silico prediction and subsequent synthesis of F317L T-cell epitopes, successfully identifiing an epitope using the IFN-γ ELISpot method. Our findings in this study will provide a basis for further in-depth studies of ASFV F317L immunogenicity, as well as a technical reference for the subsequent functional studies of the F317L protein and identification of T-cell active epitopes.

## 2. Materials and Methods

### 2.1. Protein and Cells

ASFV F317L and p72 recombinant proteins were prepared and preserved in this experiment. Experimentally obtained mice splenic lymphocytes were cultured with RPMI 1640 medium (Gibco, Grand Island, NE, USA, Cat. No#. 11875093) containing 10% Fetal Bovine Serum (FBS) (Gibco, Grand Island, NE, USA, Cat. No#. A31604-01) and 100 units/mL penicillin-streptomycin (Gibco, Grand Island, NE, USA, Cat. No#. 15140-122).

### 2.2. Animals

A total of 18 SPF (Specific pathogen Free) grade 6–8-weeks-old female C57BL/6J mice were purchased from Beijing Viton Lihua Laboratory Animal Technology Co., Ltd. (Beijing, China) and kept in a negative-pressure barrier environment in the Laboratory Animal Center of the Beijing Institute of Animal Husbandry and Veterinary Research, Chinese Academy of Agricultural Sciences. All experimental activities were approved by the Experimental Animal Welfare Ethics Committee of Beijing Institute of Animal Science and Veterinary Medicine, Chinese Academy of Agricultural Sciences (No. IAS2022-157).

### 2.3. Animal Experiments

To assess the immunogenicity of the ASFV F317L protein in mice, female SPF C57BL/6J mice were immunized with recombinant F317L protein that was previously expressed and identified. In detail, six mice in each group were subcutaneously immunized at a dose of 10 μg with prime and booster immunizations at Day 0 and Day 21, respectively. As the immunized group, positive and negative controls, mice were administered with F317L protein, p72 protein, and phosphate-buffered saline (PBS), respectively. Subsequently, serum was collected from the orbits of mice on Days 7, 14, 21, and 28, and the spleens of mice were extracted 7 days after the booster immunization (Figure 1). The splenic lymphocytes obtained from three mice in the F317L protein group were used for IFN-γ ELISpot Assays and flow cytometry assay, while another three mice were used for peptide screening.

### 2.4. Mice Splenic Lymphocytes Isolation

Mice splenic lymphocytes were isolated at Day 28 post-immunization. After being euthanized, the spleens from mice of each group (F317L protein immunized group, p72 protein positive and PBS negative controls) were taken out to make splenocyte suspension, and mice splenic lymphocytes were isolated by a Mouse Splenic Lymphocyte Isolation Kit (Solarbio, Beijing, China, Cat. No#. P8860). The middle milky white lymphocyte layer was washed twice and centrifuged 5 min under 1000 rpm; then, the splenic lymphocytes were suspended in RPMI1640 complete culture medium.

### 2.5. IFN-γ ELISpot Assays

In order to detect the cellular immunity level of spleen cells in immunized mice, IFN-γ ELISpot assays were performed using a Mouse IFN-γ precoated ELISpot kit (Dakewe, Shenzhen, China, Cat. No#2210006) according to the manufacturer’s instructions. Isolated splenic lymphocytes from mice immunized with F317L protein were seeded into pre-coated 96-well IFN-γ ELISpot plates at a density of 2 × 10^6^/mL per well, three replicate per mice, then stimulated with 100 ng of F317L protein. At the same time, the splenic lymphocytes from p72 protein immunized mice used as positive control and stimulated with 100 ng of p72 protein, and those obtained from mice received no stimulation used as negative control. All positive controls in each group were used with 5 ng PMA and 100 ng Ionomycin of positive stimulant (Dakewe, Shenzhen, China, Cat. No#2030421). Then, the plates were incubated in a culture incubator at 37 °C, 5% CO_2_ for 24 h, after which the cells were lysed for 10 min with pre-cooled deionized water at 4 °C and washed with washing buffer six times, Biotinylated Antibody was added, and plates were subsequently incubated at 37 °C for 1 h, then washed with washing buffer six times. Streptavidin-HRP was added and incubated at 37 °C for 1 h and washed with washing buffer six times. AEC color development solutions were added, kept in the dark or 30 min at room temperature (RT), and then washed with washing buffer six times. Finally, the plates were dried naturally at RT, and the spot-forming Cells (SFCs) were counted using an IRIS Mabtech ELISpot/FluoroSpot plate reader (Mabtech, Stockholm, Sweden).

### 2.6. CD4^+^/CD8^+^ T Cell Subclass Ratio

Splenic lymphocyte CD4^+^/CD8^+^ subclass analysis was performed using flow cytometry. An amount of 97 µL of cell staining buffer (BioLegend, San Diego, CA, USA, Cat. No#420201) and 0.5 µL of TruStain FcXTM PLUS (anti-mouse CD16/32) (BioLegend, Cat. No#156604) antibody were transferred to 1.5 mL Eppendorf (EP) tubes containing 1.5 × 10^6^ cells, mixed softly, and then used to stain the cells for 10 min at RT in the dark. Thereafter, 0.5 µL of FITC anti-mouse CD3ε antibody (BioLegend, San Diego, CA, USA, Cat. No#100204), 1.25 µL of PerCP/Cyanine5.5 anti-mouse CD8α antibody (BioLegend, San Diego, CA, USA, Cat. No#100734), and 1.25 µL of PE anti-mouse CD4 antibody (BioLegend, San Diego, CA, USA, Cat. No#100512) were added to the EP tubes, mixed softly, and then used to stain the cells for 30 min at 4 °C in the dark. Then, the EP tubes were centrifuged at 200× *g* for 5 min, and the supernatant was discarded. Next, 1 mL of PBS was added to make a cell suspension, centrifuged at 4 °C, 200× *g* for 5 min, and the supernatant was then removed. Cells were centrifuged twice, and 250 µL of Cell Staining Buffer was added for cell suspension. Finally, the cells were analyzed by FACSVerse flow cytometer (BD Bioscience, Franklin Lakes, NJ, USA), and all dates were analyzed using FlowJoV_10 software.

### 2.7. Indirect ELISA

The F317L and p72 protein were diluted to 2 µg/mL with carbonate buffer solution (pH 9.6). An amount of 100 µL aliquots were added to each well in 96-well ELISA plates and incubated for 14 h overnight at 4 °C. The plates were sealed with 300 µL of 5% skimmed milk at 37 °C for 2 h, then washed with PBS with 0.01% Tween-20 (PBST), and the serum (1:3200) was then added and incubated at 37 °C for 1 h. After washing with PBST, HRP-conjugated goat anti-mouse IgG (1:5000) (Abcam, Cambridge, UK, Cat. No#ab6789) was added and incubated at 37 °C for 1 h. The plates were incubated with 100 µL KPL SureBlue^TM^ TMB Microwell Peroxidase Substrate (KPL, Milford, MA, USA, Cat. No#5120-0077) at RT for 15 min, and 100 µL of 1 M HCl was added to terminate the reaction, after which the results were expressed as optical density (OD) at 450 nm. Values were assessed based on a threshold indirect ELISA value optimized in the laboratory. Samples with an experimental well/blank well ratio OD_450_ value ≥ 2.1 specifically were deemed to be positive, whereas those with a value < 2.1 were assessed to be negative.

### 2.8. Cytokine Assays

The levels of IL2, IL4, IFN-γ, and TNF-α in mouse sera were determined using a Q-Plex™ Mouse HS kit (Cayman Chemical, Michigan, USA, Cat, No#39970) according to the manufacturer’s instructions. Briefly, 30 µL of serum samples were added to the wells of plates, and the plates were covered with a plate seal and incubated at RT (20–25 °C). Thereafter, the wells were washed three times, followed by the sequential addition of 50 µL of Detection Mix, Streptavidin-HRP, and substrate with incubations at RT for 2 h, 20 min, and 5 min, respectively. The plates were then placed in the Q-View Imager (Quansys Biosciences, Utah, USA), with the images thus obtained being processed using Q-View software 3.0, resulting in the generation of charts, concentrations, and statistics.

### 2.9. Prediction of T Cell Epitopes of ASFV F317L Protein

The T cell antigenic epitope region of the F317L protein amino acid sequence (GenBank ID: UFD97811.1, Beijing, China) was analyzed by IEDB server V2.22 (https://www.iedb.org/, accessed on 4 September 2023) and NetMHCpan-4.1 server (https://services.healthtech.dtu.dk/services/NetMHCpan-4.1, accessed on 4 September 2023). Peptide affinity was assessed based on a percentile rank, with a lower rank% value being taken to be indicative of a stronger peptide affinity. The combined results yielded 29 high-affinity peptides (Table 1), the predicted sequences of which were chemically synthesized by GenScript Ltd. (Nanjing, China). These peptides, for which the purity was greater than 95%, were dissolved in the recommended solvents (ultrapure water, formic acid, DMSO, paminoacetic acid) at a concentration of 2 mg/mL.

### 2.10. F317L Protein T Cell Epitope Screening

The splenic lymphocytes of immunized mice were used for T-cell epitopes screening, and were stimulated by the epitopes with 10 μg per well. The negative control was stimulated with solvents, and the positive control was stimulated with 5 ng PMA and 100 ng Ionomycin of positive stimulant. All the treated cells were incubated at 37 °C, 5% CO_2_ for 20 h. Then, the following steps processed as described in Section 2.6.

### 2.11. Statistical Analysis

Statistical analysis and plotting were performed using GraphPad Prism 8.0 software. Comparisons between groups were performed using a student’s *t*-test. Data were expressed as mean ± standard deviation (SD). The *p*-value obtained from each analysis was indicated in the graph, and *p*-values < 0.05 were considered statistically significant.

## 3. Results

### 3.1. The F317L Protein Stimulates Splenic Lymphocytes to Secrete IFN-γ

IFN-γ ELISpot assays were performed on Day 7 after booster immunization to detect the cellular immunity level of the spleen cells of immunized mice. The results revealed that the splenic lymphocytes of F317L protein and p72 protein in immunized mice produced higher levels IFN-γ than those in the negative control group (*p* < 0.001) when stimulated with 100 ng of F317L protein and p72 protein, respectively (Figure 2). The numbers of IFN-γ spots stimulated with the F317L protein were significantly higher than those in the p72 positive control group (*p* < 0.01). These findings accordingly indicated that the F317L protein could effectively activate the cellular immunity and stimulate the secretion of IFN-γ by the spleen lymphocytes of immunized mice.

### 3.2. The F317L Protein Can Induce an Increase in the CD4^+^/CD8^+^ Ratio of Mouse Splenic Lymphocytes and Promote Differentiation to CD4^+^ T Cell Subsets

Flow cytometry analysis of T lymphocyte subpopulations in each group of mice revealed significant differences between the F317L protein group and the negative (*p* < 0.01) and/or p72 positive control groups (*p* < 0.05) with respect to the CD4^+^/CD8^+^ ratio of splenic lymphocytes (Figure 3), which indicated that the F317L protein could stimulate the splenic lymphocytes of mice to differentiate toward CD4^+^ T cells and activate immune response in vivo.

### 3.3. The F317L Protein Induced and the Secretion of High Specific Antibodies in Humoral Immunity in Immunized Mice

Indirect ELISA assays were performed to detect specific antibody levels on Days 7, 14, 21, and 28 post-immunization. The results shows that specific antibodies were detected as soon as Day 7, and the antibody levels increased gradually with time, being maintained at an elevated level during the remaining period (Figure 4). Additionally, the specific antibody levels in the F317L protein group were lower than those in the p72 positive control group, although they still remained at a high level. These findings indicated that the F317L protein could induce the humoral immunity.

### 3.4. F317L Protein Promoted the Secretion of Immune-Related Cytokines (IL-2, IL-4, IFN-γ, and TNF-α) in Immunized Mice

The serum cytokine contents measured on Day 21 after primer immunization and 7 days after the booster immunization revealed that the levels of IL2, IL4, IFN-γ, and TNF-α cytokines in F317L protein immunized mice were higher than those in the negative control group mice (Figure 5). Additionally, the level of Day 28 post-inoculation was higher than Day 21 (*p* < 0.001). The results indicated that F317L protein promoted the secretion of immune-related cytokines (IL-2, IL-4, IFN-γ, and TNF-α) in immunized mice both in the initial and booster immunization stage, and the level of cytokines produced after secondary immunization was higher than the primer, with the difference being very significant (*p* < 0.001).

### 3.5. Peptide^(246−255aa)^ Is Potential T Cell Active Epitopes of F317L Protein

To identify the T cell epitope of the F317L protein, we predicted and subsequently synthesized 29 peptides of 9~11 amino acids, which were subjected to IFN-γ ELISpot analyses by co-incubating cells exposed to different stimuli. The results obtained on the 7th day following booster immunization with the F317L protein revealed different numbers of IFN-γ spots in cells treated with the different peptides. In particular, the number of spots per SFC/10^5^ cells treated with the F25 (^246^SRRSLVNPWT^255^) peptide was found to be significantly higher than that of the F317L protein (Figure 6). Accordingly, F25 (^246^SRRSLVNPWT^255^) peptide was identified as a putative T-cell epitope by IFN-γ ELISpot assays, which can induce a strong immune response.

## 4. Discussion

Despite the fact that African swine fever has been endemic for over a century, to date, no effective and safe vaccine against it has been successfully developed. ASFV is characterized by a large genome containing between 160 and 175 open reading frames [19] that encode numerous proteins that contribute to the initiation of complex immune escape mechanisms [5]. Although inactivated vaccines produce strong humoral immunity, they tend to be ineffective in stimulating cellular immune responses, and therefore fail to provide immune protection [20,21,22]. However, vector-based vaccines constructed using multiple types of viral antigens (including p220, p72, p15, B602L, p62, p32, p54, EP153R, p12, and CD2v [23,24,25,26]) that produce good specific antibody responses, and a certain degree of cellular immunity, can provide partial protection [9,27,28,29,30]. Moreover, safety remains a major concern for weak vaccines [31], and, although gene deletion-based vaccines can provide partial protection [32], there is a need to identify suitable target genes and combinations of deletions [33]. Currently, the ASF subunit vaccine can produce better humoral immunity [34] but cannot stimulate efficient cellular immunity. Consequently, identifying effective protective antigens and establishing approaches for activating cellular immunity [10,31] are of particular importance for vaccine development [16].

The results in this study provided evidence to indicate that F317L protein may serve as a potential immunogenic antigen of ASFV. Given that the ASFV p72 protein is a major component of the viral capsid, which is an important criterion for virus typing [35] and also an important antigen of ASFV, we used this protein as a positive control in this study. The results of indirect ELISA analyses indicated that the F317L protein can activate humoral immunity, which can stimulate mice to produce large numbers of specific antibodies. In addition, we used the sensitive ELISpot assay to detect IFN-γ [36], which can serve as an important indicator for assessing the activation of cellular immunity. The results indicated that the F317L protein can significantly stimulate the production of IFN-γ and activated cellular immune responses in F317L protein immunized mice compared to the p72 protein positive control. In addition, flow cytometry analyses revealed a significant increase in the CD4^+^ T percentage of T lymphocytes stimulated by the F317L protein compared with that of p72 protein positive control. Cytokines are important mediators of the immune response, with Th1 cells mainly secreting IL-2, IFN-γ, and TNF-α, and Th2 cells mainly secreting IL-4. Cytotoxic T lymphocytes (CTLs) can play important roles in combatting infection via the secretion of cytokines such as IFN-γ and TNF-α [37], the former of which, secreted by cells in response to viral infection, can enhance natural killer (NK) cell and macrophage activity, while porcine IFN-γ has been established to reduce ASFV replication in porcine macrophages in vitro [38]. The results of our cytokine assays in this study revealed significantly elevated levels of immune-related cytokines (IL-2, IL-4, IFN-γ, and TNF-α) in response booster immunization, which also provides evidence to indicate that these immunized mice produced both humoral and cellular immune responses.

Studies in recent years have shown that ASFV-neutralizing antibodies alone do not provide adequate protection from infection [9,21,39]. Both NK and CTL cells detect and attack pathogen-infected cells and play important roles in mounting a resistance to ASFV infection [40,41]. The findings of some studies have indicated that the CD8α^+^ T lymphocyte response is closely associated with immunity protection [10,15,33,42], providing evidence that cellular immunity is a key facet of the host response to ASFV infection. Cellular immunity is mediated primarily via an endogenous delivery pathway, which implies that active T cell epitopes with high affinity for SLA (swine leukocyte antigen) molecules play an essential role in activating cellular immune responses. In this regard, Sun et al. [43] coupled multiple T cell epitopes of ASFV and constructed a nano-vaccine, which was found to be highly immunogenic and produced strong humoral and cellular immune responses in mice following immunization. Given that antigenic peptides can bind directly to SLA molecules and efficiently stimulate strong humoral and cellular immunity, the screening and characterization of antigenic peptide libraries have attracted considerable attention in recent years. Seventeen peptides derived from 15 different antigens screened using SLAI were found to stimulate strong humoral and cellular immune responses in pigs immunized for 21 days [44]. Lymphocyte peptide libraries for anti-ASFV immunization were screened using surviving porcine peripheral blood mononuclear cells (PBMCs) and randomly divided into sub-peptide Libraries A and B for immunization, with both groups being found to stimulate the production of strong cellular immune responses [10]. The T cell epitope of the CD2V single antigen, which promotes CD8^+^ T cell metabolism, was found to induce significantly greater viral clearance than whole-protein immunization [45].

The findings of these studies have highlighted that ASFV specific T cell epitopes are important factors in stimulating the generation of efficient cellular immunity, and, consequently, the prediction and screening of T cell epitopes are considered key steps in the development of novel subunit vaccines. The techniques currently employed for predicting and screening T cell epitopes are relatively well developed, with in silico prediction, IFN-γ ELISpot, and immunepeptidomics approaches being commonly adopted [16,36,46]. In the present study, we used relevant software (IEDB server V2.22 and NetMHCpan-4.1 server) to predict and subsequently synthesize 29 F317L protein T-cell epitopes, and we also screened for active T cell epitopes using an IFN-γ ELISpot assay. Among these, we demonstrated that peptide F25 (^246^SRRSLVNPWT^255^) could stimulate mouse splenic lymphocytes to produce large amounts of IFN-γ, and, in this regard, it proved to be more potent than the stimulatory effects of the full-length F317L protein. We firstly and systematically evaluated the humoral immunity and cellular immunity of ASFV F317L protein, further identifying the T cell epitope of this protein, and the results thus provide evidence to indicate that the F25 (^246^SRRSLVNPWT^255^) peptide is a putative T cell-dominant epitope. Given that data obtained using pigs can provide a more reliable indication of the immunogenicity of ASFV proteins, in further studies we will seek to validate the effects of this F317L protein T cell epitope on surviving porcine PBMCs, which could accordingly provide a basis for the development of a novel ASF subunit vaccine based on a peptide library of antigenic epitopes, with the aim being to provide effective protection.

## 5. Conclusions

In this study, we conducted animal experiments to assess the immunogenicity of the African swine fever virus F317L protein, which was found to stimulate both humoral and cellular immune responses in mice. In addition, by performing IFN-γ ELISpot analyses, we identified a potentially active T cell epitope F25 (^246^SRRSLVNPWT^255^). Collectively, our findings indicate that the F317L protein has good immunogenicity in mice. Further studies will focus on evaluating the immunogenicity of this protein and the elicited immune protection in pigs, the findings of which will provide a reference for the screening of target antigens of African swine fever vaccines.

## Figures and Tables

**Figure 1 animals-14-01331-f001:**
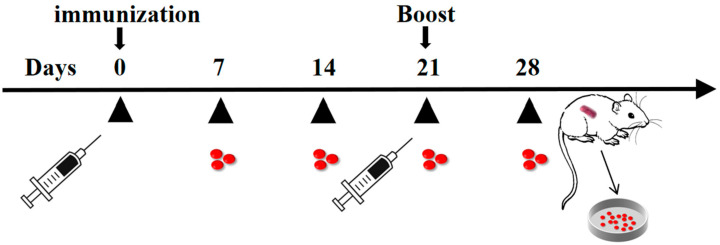
Experimental procedures of the immunization and sample collection. The first immunization was performed on Day 0, with a booster immunization at Day 21, and serums were collected on Days 7, 14, 21, and 28, while splenic lymphocytes were collected on Day 28. Red dots represent serum.

**Figure 2 animals-14-01331-f002:**
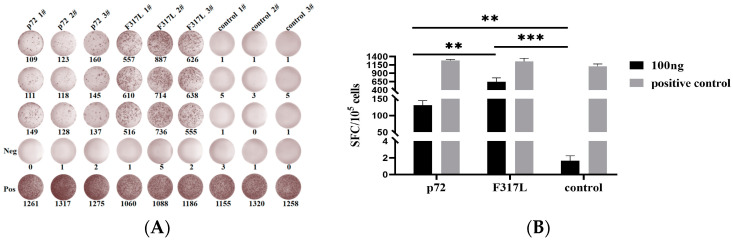
The IFN-γ secreted by splenic lymphocytes of immunized mice detected by IFN-γ ELISpot assays. (**A**) IFN-γ ELISpot spot plots. Splenic lymphocytes stimulated with 100 ng/well of F317L protein and p72 protein. (**B**) Histogram of spots counted in IFN-γ ELISpot assays. Data are expressed as mean ± SD. According to the number of spots counted in IFN-γ ELISpot assays, the data are representative of the mean ± SD calculated based on three mice and three parallel holes in each mouse. The difference between the F317L group compared to the negative control group (*p* < 0.001) and the p72 positive control group (*p* < 0.01) was statistically significant. Note: ** *p* < 0.01; *** *p* < 0.001.

**Figure 3 animals-14-01331-f003:**
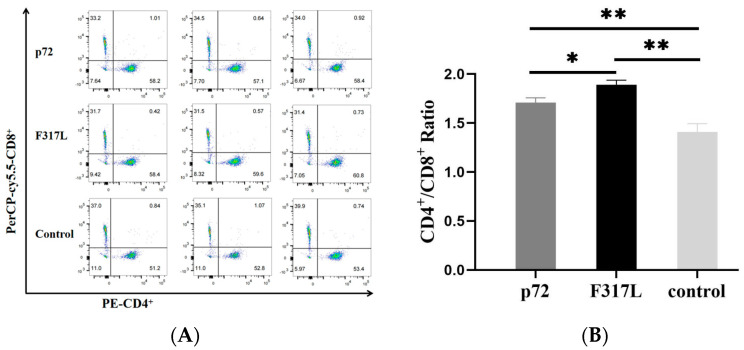
Results of T cell subpopulation assays. (**A**) Flow cytometry results for each immunization group. (**B**) Histogram of CD4^+^/CD8^+^ ratio. Statistical analysis of the T cell subset ratios showed significant differences in the F317L protein group compared with the negative control group (*p* < 0.01) and the p72 positive control group (*p* < 0.05). Note: * *p* < 0.05; ** *p* < 0.01.

**Figure 4 animals-14-01331-f004:**
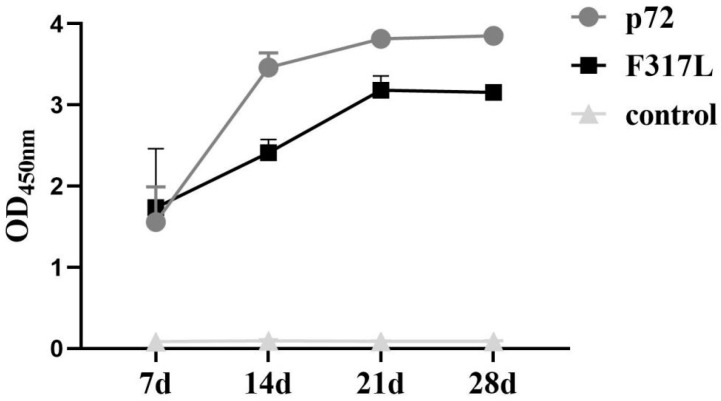
Specific antibody levels in humoral immunity in immunized mice. The levels of specific antibodies in the serum of 7 d, 14 d, 21 d, and 28 d in each group were detected, and the line graphs of specific antibodies were plotted.

**Figure 5 animals-14-01331-f005:**
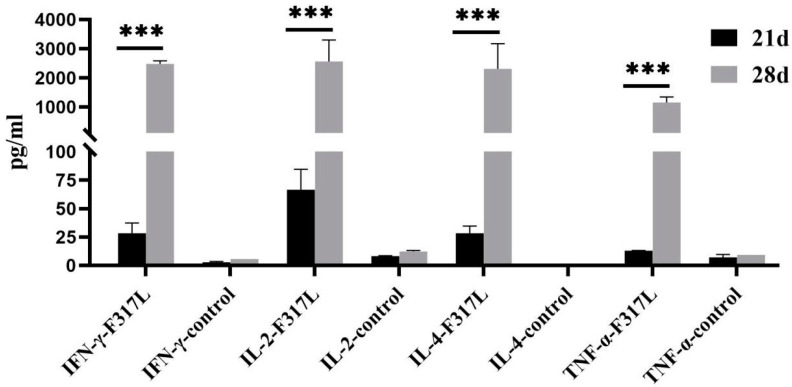
The contents of cytokines IL2, IL4, IFN-γ, and TNF-α in the serum of mice on 21 and 28 days. Notably, 21 d represents the serum sample on the 21st day after the first immunization, while 28 d represents the serum sample on the 7th day after the second strengthened immunization. It is statistically significant. Note: *** *p* < 0.001.

**Figure 6 animals-14-01331-f006:**
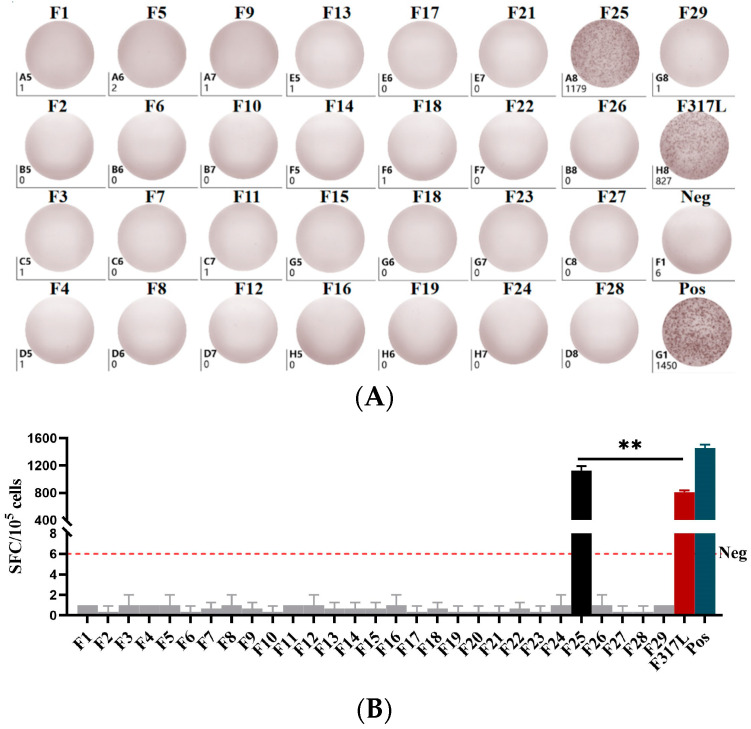
Results of IFN-γ ELISpot experiments. (**A**) F317L 29 peptide, protein, negative and positive control IFN-γ ELISpot spot plots (only one mouse spot plot is shown). (**B**) Histogram of IFN-γ ELISpot experiment results. With positive control established, the results showed that peptide F25 (^246^SRRSLVNPWT^255^) had a statistically significantly higher number of spots per SFC/10^5^ cells than F317L protein (*p* < 0.01). Note: ** *p* < 0.01.

**Table 1 animals-14-01331-t001:** Sequence information table of predicted and synthesized protein peptides.

Number	Peptide	Number	Peptide	Number	Peptide
F1	^1^MVETQMDKL^9^	F11	^115^DDDREWCGR^119^	F21	^178^CGGMPSICD^186^
F2	^26^SNAHITQTM^34^	F12	^117^CGRINMING^125^	F22	^204^IISSNQVGM^212^
F3	^40^ENHSVDGGA^48^	F13	^123^INGVPEIVE^131^	F23	^211^GMLTVDKRII^220^
F4	^46^GGAAKNVSK^54^	F14	^129^IVEIIPSPY^137^	F24	^219^IIVDLWANEN^228^
F5	^53^SKGKSSPKE^61^	F15	^137^YRAGENIYF^145^	F25	^246^SRRSLVNPWT^255^
F6	^57^SSPKEKKHW^65^	F16	^148^EAMMPADIY^156^	F26	^262^ILQDYGIEY^270^
F7	^60^KEKKHWTEF^68^	F17	^157^SRVANKPAMF^166^	F27	^270^YIIFPSNDF^278^
F8	^63^KHWTEFESW^71^	F18	^160^ANKPAMFVF^168^	F28	^276^NDFFIYEDE^284^
F9	^73^QLSKSKRSF^81^	F19	^166^FVFHTHPNL^174^	F29	^294^TNFFTLHEL^302^
F10	^76^KSKRSFKEY^84^	F20	^172^PNLGSCCGG^180^		

## Data Availability

No new data were created.

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
