# Peer review of "Analysis of the Immunogenicity of African Swine Fever F317L Protein and Screening of T Cell Epitopes"

_animals, 2024, doi:10.3390/ani14091331_

Round 1

Reviewer 1 Report

Comments and Suggestions for Authors

Dear editor, 

The manuscript provided by Dr. Huang et al., is an interesting paper, because it represents the work that I consider to be the backbone for understanding ASFV replication and virulence, and plays an important role in future development of ASF vaccine. 

The manuscript describes the immunogenicity of ASFV F317L protein and its epitopes that have a potential to activate T-cell responce in mice. 

As I started reading the manuscript, I couldn't understand the results or the discussion section, because in my opinion the paper needs a full english review. So I advice the authors to submit the paper to english proofreading either at MDPI or other independent agencies. 
The materials and methods are not clear to me, and I got lost in them. 

Therefore, I am unable to revise the paper before being improved and edited for english. Then I think we can analyze the paper.

Regards, 

Comments on the Quality of English Language

The english language of the manuscript is so poor, and I advice to submit the paper for revision by native english speakers. 

Author Response

Dear  Reviewer,

Thank you for your letter and reviewer comments on our manuscript entitled "Analysis of the immunogenicity of African swine fever F317L protein and screening of T cell epitopes" (Manuscript number: animals-2932645). We thank the reviewers for all their valuable comments on our work. We revised the manuscript based on the reviewers' comments, questions, and suggestions. The main corrections to the paper and responses to the reviewers' comments are below.

1. Comment: Therefore, I recommend that the author submit the paper to MDPI or other independent agency for English proofreading. The materials and methods are not clear to me and I am lost in them.

Response: We apologize for our English writing errors and this article has been proofread in English. Additionally, the Materials and Methods section has been revised. The proteins related to this experiment were pre-prepared and preserved by the laboratory, therefore, in this study, the relevant protein preparation process is not provided in this article.

Reviewer 2 Report

Comments and Suggestions for Authors

The manuscript entitled "Analysis of the immunogenicity of African swine fever F317L protein and screening of T cell epitopes” contains interesting but some improvements

Main

For such a study, it is important to compare the immunogenicity of the protein being tested with any protein whose immunogenicity is known. If such a study has not been conducted, it is necessary to justify the relevance of such work. There are many studies in the scientific literature on the immunogenicity of other ASF virus proteins; what is the benefit of the F317L immunogenicity study? The authors claim that their study of viral proteins "they all failed to provide ideal immune protection", does this mean that they are confident that F317L will help provide ideal immune protection?

For such studies, the use of a positive control (with a well-known antigen) for immunization of mice and for understanding the synthesis of cytokines, and especially the interferon action, is very important. The authors ignore this.

Figure 4 without positive control is meaningless.

Minor

Some figures need to be enlarged (Figure 2A).

In the legends to the figures (Fig. 2, 3) various degrees of confidence are mentioned (* p<0.05; ** p<0.01; *** p<0.001.), however, there is only one confidence level in the graphs.

The term "excellent immunogenicity" - compared to what?

In my version of the article in Table 1 there are problems in numbering.

Author Response

Dear  Reviewer,

     Thank you for your letter and for the reviewers’ comments concerning our manuscript entitled “Analysis of the immunogenicity of African swine fever F317L protein and screening of T cell epitopes” (Manuscript ID: animals-2932645). We appreciate all of the valuable comments from the reviewers of our work. We have revised our manuscript according to the reviewer’s comments, questions and suggestions. The main corrections in the paper and the responds to the reviewer’s comments are as following.

1.Comment: For such a study, it is important to compare the immunogenicity of the protein being tested with any protein whose immunogenicity is known. If such a study has not been conducted, it is necessary to justify the relevance of such work.

Response: We appreciate it very much for this good suggestion, and we agree that positive control group would be useful to evaluate immunogenicity of F317L protein. We chose the p72 protein as a positive control group and supplemented this relevant experimental data at line 220/235/248.

2.Comment: There are many studies in the scientific literature on the immunogenicity of other ASF virus proteins; what is the benefit of the F317L immunogenicity study?

Response: So far, studies on the immunogenicity of the F317L protein have not been reported, and the study of its immunogenicity in this paper may provide a new target antigen choice for ASF vaccine.

3.Comment: The authors claim that their study of viral proteins “they all failed to provide ideal immune protection”, does this mean that they are confident that F317L will help provide ideal immune protection?

Response: We did not perform protective studies of the F317L protein in this study, so the description of this paper about this place has been modified.

4.Comment: For such studies, the use of a positive control (with a well-known antigen) for immunization of mice and for understanding the synthesis of cytokines, and especially the interferon action, is very important. The authors ignore this.

Response: We have re-written this part according to the Reviewer’s suggestion at line 311-318.

5.Comment: Figure 4 without positive control is meaningless.

Response: Figure 4 A positive control has been added at line 248.

6.Comment: Some figures need to be enlarged (Figure 2A).

Response: Figure 2A figures have been enlarged at line 219.

7.Comment: In the legends to the figures (Fig. 2, 3) various degrees of confidence are mentioned (* p<0.05; ** p<0.01; *** p<0.001.), however, there is only one confidence level in the graphs.

Response: We are very sorry for our negligence and have made the corresponding changes at line 226 and 238.

8.Comment: The term “excellent immunogenicity” – compared to what?

Response: The F317L has excellent immunogenicity compared to p72 protein.

9.Comment:  In my version of the article in Table 1 there are problems in numbering

Response: The numbering of table 1 has been changed.

Reviewer 3 Report

Comments and Suggestions for Authors

African swine fever virus (ASFV) infection is causing significant economic losses to the global pig industry, making its control an urgent issue. F317L, one of the ASFV proteins, has an immunosuppressive effect on the host. In this study, the authors showed strong humoral and cell-mediated immune responses in mice. Additionally, they identified a T cell epitope for F317L as compared full F317L protein. Overall, this paper is interesting and future developments in ASFV epidemic prevention are highly anticipated.

Major point

Nothing

Minor points

1. Line 22; Isn’t “did’t been known” “have not been known”?

2. Line 43; There are duplicate periods after [2], so delete one.

3. Lines 104-105; Leave one line blank.

4. Line 121; “kit” is duplicated, so delete the last one.

5. Line 122; Isn't the 6 in “2x106” a superscript?

6. Line 129; Isn't "with ()" necessary after "washed"?

7. Line 129; What is the antibody used for?

8. Lines 130-135, 146-151, 176-178, and 216; These sentences are imperative sentences. Make its regular sentences!

9. Line 143; Isn't the 7 in “1x107” a superscript?

10. Line 143; What is “EP”? Don't omit it.

11. Line 181; What is “.csv file”?

12. Line 299; I think it's better to start with IFN-γ.

13. Line 355; Delete “Lidan”.

14. Line 377; Insert a space after “)”.

15. Line 384; The word “late protein” appears at the end, so it feels a little strange. I think it should be written in the Introduction as well.

Author Response

Dear  Reviewer,

     Thank you for your letter and for the reviewers’ comments concerning our manuscript entitled “Analysis of the immunogenicity of African swine fever F317L protein and screening of T cell epitopes” (Manuscript ID: animals-2932645). We appreciate all of the valuable comments from the reviewers of our work. We have revised our manuscript according to the reviewer’s comments, questions and suggestions. The main corrections in the paper and the responds to the reviewer’s comments are as following.

1.Comments: Line 22; Isn’t “did’t been known” “have not been known”?

Response: This article has been proofread in English.

2.Comments: Line 43; There are duplicate periods after [2], so delete one.

Response: The duplicate periods has been deleted.

3.Comments: Lines 104-105; Leave one line blank.

Response: I've finished revising this place.

4.Comments: Line 121; “kit” is duplicated, so delete the last one.

Response: The duplicate “kit” has been deleted at line 126.

5.Comments: Line 122; Isn't the 6 in “2x106” a superscript?

Response: We are very sorry for our negligence and have made the corresponding changes at line 129.

6.Comments: Line 129; Isn't "with ()" necessary after "washed"?

Response: It was rectified at line 136.

7.Comments: Line 129; What is the antibody used for?

Response: Biotinylated Antibody can specifically bind IFN-γ at line 136.

8.Comments: Lines 130-135, 146-151, 176-178, and 216; These sentences are imperative sentences. Make its regular sentences!

Response: This article has been proofread in English.

9.Comments: Line 143; Isn't the 7 in “1x107” a superscript?

Response: We are very sorry for our careless mistake and it was rectified at line 148.

10.Comments: Line 143; What is “EP”? Don't omit it.

Response: The “EP” means Eppendorf tube, at line 148.

11.Comments: Line 181; What is “.csv file”?

Response: “.csv file” is a simple and practical file format for storing data for this experiment.

12.Comments: Line 299; I think it's better to start with IFN-γ.

Response: I have completed the English changes here.

13.Comments: Line 355; Delete “Lidan”.

Response: The “Lidan” has been deleted ai line 327.

14.Comments: Line 377; Insert a space after “)”.

Response: We are very sorry for our careless mistake and it was rectified.

15.Comments: Line 384; The word “late protein” appears at the end, so it feels a little strange. I think it should be written in the Introduction as well.

Response: We have made correction according to the Reviewer’s comments at line 71.

Round 2

Reviewer 2 Report

Comments and Suggestions for Authors

The shortcomings of the article indicated in the review have been generally corrected